# Effects of Different Drying Methods on Untargeted Phenolic Metabolites, and Antioxidant Activity in Chinese Cabbage (*Brassica rapa* L. subsp. chinensis) and Nightshade (*Solanum retroflexum* Dun.)

**DOI:** 10.3390/molecules25061326

**Published:** 2020-03-13

**Authors:** Millicent G. Managa, Yasmina Sultanbawa, Dharini Sivakumar

**Affiliations:** 1Phytochemical Food Network Research Group, Department of Crop Sciences, Tshwane University of Technology, Pretoria West 0001, South Africa; managagm@gmail.com; 2Australian Research Council (ARC), Queensland Alliance for Agriculture and Food Innovation, Center for Food Science and Nutrition, The University of Queensland, QLD 4108 Brisbane, Australia; y.sultanbawa@uq.edu.au

**Keywords:** Traditional African leafy vegetables, poly phenols, chlorogenic acid, kaempferol glycosides, antioxidants, postharvest processing

## Abstract

Chinese cabbage (*Brassica rapa* L. subsp. chinensis) and Nightshade (*Solanum retroflexum* are popular traditional leafy vegetables consumed predominantly by rural Africans. Sun drying is adopted as a traditional method of postharvest preservation to store theses leaves during off seasons. The influence of different types of postharvest processing treatments, such as conventional oven drying, solar cabinet drying, sun drying and freeze drying, on the changes on colour properties and antioxidant components were investigated. Freeze-drying retained the ascorbic acid content, antioxidant activities, total chlorophyll content, green colour by reducing the colour difference (∆E). With regard to Chinese cabbage and Nightshade leaves, sun and microwave drying respectively had the most negative impact on all the identified phenolic compounds. The OPLS-DA and the UPLC–QTOF/MS and chemometric approach showed kaempferol-3-*O*-sophoroside, kaempferol-3-sophorotrioside-7-glucoside and hydroxyoctadecenedioic acid as the markers responsible for the separation of sun-dried samples from the other drying treatments in Chinese cabbage. Sinapoyl malate was not detected in sun-dried samples. Caffeoylmalic acid was identified as the marker compound to separate the other drying treatments from the microwave dried samples of Nightshade leaves. Trihydroxyoctadecadiene derivative and hydroxyoctadecanedioic acid were detected in microwaved samples. Due to the cost effectiveness, solar dryer cabinet treatment was recommended for drying both vegetables. The proximate analysis of solar dried functional powder of Chinese cabbage and Nightshade vegetables demonstrated higher contents of protein and dietary fibre.

## 1. Introduction

Chinese cabbage (*Brassica rapa* L. subsp. chinensis) and Nightshade (*Solanum retroflexum* Dun) are popular vegetables among rural and peri-urban households in Southern Africa. The Ca, Fe, glucosinolates and β-carotene contents were reported as 1020 g kg^−1^ FW, 26 g 36 kg^−1^ FW and 26 g 36 kg^−1^ DW respectively, and the kempferol content varied from 0.2002 to 0.25 g kg^−1^ on dry weight basis [1,2]. In addition, the raw leaves of *Brassica rapa* L. subsp. Chinensis contain kaempferol-sophoroside-O-hexoside, kaempferol-dihexoside, kaempferol-sophoroside, kaempferol hexoside, ferulic acid and myrectin-O-arabinoside. Sinigrin was reported as the highest glucosinolate in freshly harvested Chinese cabbage [3]. The Nightshade leaves contained higher levels of Ca (199 g 100 g^−1^), Mg (92 g 100 g^−1^), and Fe (7.2 g 100 g^−1^) compared to raw spinach [4] and the following phenolic compounds, neochlorogenic, chlorogenic, and caffeoylmalic acid, kaempferol O-rhamnosyl hexoside and rutin [3].

Since, the growing alertness of functional compounds found in the fruits and vegetables and their beneficial health effects in recent years has pointed to the fact that increased inclusion of fruits and vegetables is essential in our daily diets [5]. Traditional African leafy vegetables are well known for the contribution of both micronutrients and functional compounds to the diets of African consumers [6]. Therefore, traditional African leafy vegetables can be included in diet diversification strategy for the sub-Saharan African population to combat the hidden hunger [7]. Generally African leafy vegetables are consumed as soups, or relishes as side dishes with the main meal the carbohydrate staples. Research reports have shown that African vegetables are an abundant source of non-nutritive phytochemicals and their biological activities have been associated with many health benefits, such as anti-diabetic or obesity effects [8]. For example, moringa leaves (*Moringa oleifera*), rich in flavonoid aglycons quercetin and kaempferol in methanolic extract, showed a significant antidiabetic and antioxidant activity [8]. Leaf extracts of *Amaranthus spinosus* demonstrated anti-tumour properties opposing liver, breast and colorectal cancer cells [9].

Production of African leafy vegetables in Southern African region is encouraged at home garden or commercial level because they contribute positively towards food production and security due to their draught tolerance [5]. The highly perishable nature of the African leafy vegetables limits their marketability therefore most are dried in shade or under sun as a method adopted for preservation during the off season to remove moisture from the vegetable and to increase shelf life by preventing microbial decay without adding any preservatives [10]. The postharvest drying process and the type of methods adopted can result in changes in functional compounds and physical properties of green leafy vegetables [11]. Furthermore, postharvest drying provides an opportunity to earn a higher income from traditional vegetable functional food. The global functional foods market size is estimated to increase in 2025 to US $275.77, specifically due to the increasing consumer demand for nutritional and fortifying food additives [12]. *Amaranthus hybridus*, dried at 40 °C for 12 h, increased fibre content to 7% and β carotene and vitamin C were decreased at levels of 19.4% and 13.9% respectively [13]. Traditionally adopted sun drying has many drawbacks mainly because it is difficult to control large quantities, to achieve homogenous quality standards and to implement food safety guidelines; also, little is known about the impact of different drying methods on the phenolic compounds, antioxidant activity and ant nutritive compounds in Chinese cabbage and Nightshade leafy vegetables. However, the type of drying method adopted to dry the vegetables for the use of functional ingredients for the development of value-added product must be standardised. 

Consequently, the objective of this study was to instigate the impact of different drying methods, such as sun drying, shade drying, drying in a solar cabinet dryer, hot air oven and freeze drying methods, on the changes in i) colour properties, ii) changes in lipophilic pigments, iii) phenolic metabolite components, iv) ascorbic acid and v) antioxidant property in Chinese cabbage and Nightshade leafy vegetables.

## 2. Results and Discussion

### 2.1. Colour Changes and Lipophilic Pigments 

Table 1A,B illustrate the colour changes that took place during different types of postharvest drying in Chinese cabbage and Nightshade leaves respectively. The following colour coordinates were used to express the colour changes. Light intensity (*L**), colour coordinate *a** relates to red and green, colour coordinate *b** relates to yellow colour*, h°* = arc tangent b*/a* represents a basic colour like red (~29°), orange (~45°) or yellow (~70*°), ∆E*-colour difference, the variation in colour compared to the original sample using the following formulae given in Section 3.4.

Freeze-dried powers of both types of leaves showed significantly highest *L** (luminosity), *b**, *h°* values and the lowest *a** value, whilst the oven drying showed the opposite effect by showing significantly reduced *L*, h°* values and highest *a** value (Table 1A,B). 

On the contrary during hot air (oven) and freeze drying of cabbages (*Brassica oleracea* L. variety Capitata L.), no differences in brightness (*L**) value was reported [14].However, the increase in *b** value during postharvest dehydration treatments compared to the fresh leaves observed in Chinese cabbage (Table 1A) and Night shade leaves (Table 1B) are possibly due to the interference of intercellular air trapped on the greenness of the chlorophyll as explained previously by Rajkumar et al., [14] and confirms their findings.

The purchasing power of consumers is determined by the colour (quality) of the products [15]. Thus, the ΔE value plays a crucial role in relating exact colour changes in dried products; if *ΔE* is less than 1.0 it means the colour differences cannot be noticeable [16]. In this study, Chinese cabbage and Nightshade leaves subjected to different drying treatments exceeded the *ΔE* value 1.0, and the *ΔE* value of the freeze-dried samples showed the lowest (Chinese cabbage, *ΔE* = 2.79; Night shade *ΔE* = 3.18) values than the samples from other drying treatments. Conversely, the samples from the oven drying method showed the highest *ΔE* value (Chinese cabbage, *ΔE* = 25.93; Nightshade *ΔE* = 14.66). The greater *ΔE* values indicated lighter leaves than the fresh and freeze-dried samples. 

The influence of different drying treatments on Chinese cabbage and Nightshade leaves are given in Table 2A,B. The total chlorophyll and carotenoid content were significantly higher in all dried leaf samples compared with the fresh samples. A similar increase in lipophilic pigments during heat treatment was noted in *Ipomoea aquatica Forsk* [17] and mustard, mint and spinach greens [18]. The influence of drying treatment on Chl a, and Chl b affected the total Chl content. In freeze-drying, the non-thermal treatment influenced the retention of total chlorophyll and carotenoid contents in both vegetables, similar to the findings of Shin et al., [17]. The total chlorophyll content in freeze-dried Chinese cabbage and Nightshade leaves showed a 3.11- and 2.38-fold increase respectively, compared to the fresh leaf samples (Table 2A,B). In contrast, the oven drying degraded both the total chlorophyll and carotenoid contents in Chinese cabbage leaves by 3.03% and 69.61% respectively. Similarly, oven drying degraded the chlorophyll in Nightshade leaves by 69.66% and carotenoids by 68.29%. Previous research findings of freeze-drying (lyophilisation) confirm the findings of this study on the preservation of the carotenoid content in leafy colour of the final product is associated with visual appearance and the consumer acceptance during marketing [19,20]. During oven drying, the higher temperature (100 °C) most probably converted the chlorophyll to pheophytin and as a result, the leaves turned olive green in colour [20]., which was reflected by the higher colour change *∆E* in Table 1A,B The differences observed in the *ΔE* value can probably be attributed due to the applied drying treatment temperature, duration of the treatments influencing the removal of water molecules, resulting in changes in structural, textural changes and dry matter content [20]. Previous research further confirmed that freeze-drying was the most appropriate drying method to retain the colour of the vegetables and fruits. Freeze-drying provides a porous structure to the product, with limited or less shrinkage to preserve the product quality [21]. 

### 2.2. Changes in Non-Targeted Phenolic Metabolites.

Total ion chromatograms of untargeted metabolites of Chinese cabbage and Nightshade leaves subjected to different drying treatments in the ESI- mode by UPLC–QTOF/MS were shown in Appendix A. The traditional sun drying method was adopted as the control method to compare the changes in the non-targeted phenolic metabolite profile.

Tentative identification of 13 compounds in Chinese cabbage and 11 compounds in Nightshade leaves were illustrated in Table 3A,B. In Chinese cabbage, 53.8% of the compounds were kaempferol derivatives, whilst in Nightshade leaves, kaempferol derivatives represented only 9.0%. This clearly showed that Chinese cabbage leaves are rich source of kaempferol derivatives. In Nightshade leaves, the majority (45%) of identified compounds were phenolic acids and 27% were tigogenin with sugar molecules attached. The UPLC–QTOF/MS analysis helped to identify the kaempferol derivatives in Chinese cabbage leaves as kaempferol-3-*O*-sophoroside 7-*O*-glucoside (Compound 3; [(M−H)^−^ at *m*/*z* 191.095 and MSE fragments at *m*/*z* 609,285,255, kaempferol in combination with glucose and acylated with hydroxycinnamic acid (sinapic, and ferulic acid) derivatives such as kaempferol-3-*O*-hydroxyferoyl-trihexoside (Compound 4; with [(M−H)^−^ at *m*/*z* 963,2401 and MSE fragments at *m/z* 801,609,285), kaempferol-3-*O*-hydroxyferoyl-diglucoside (compound 9; [(M−H)^−^ at *m*/*z* 801.1874 and MSE fragments at *m*/*z* 771,609, 285,255], kaempferol-di-hexoside. kaempferol-dihexoside (Compounds 5; [(M–H)^−^ at *m/z* 609.1474 and MSE fragments at *m*/*z* 447,285), and acylated kaempferol hexoside glycosylated with sophorotrioses (sinapoyl-acylated derivatives) kaempferol-3-*O*-sinapoyldihexoside-hexoside (compound 6; [(M−H)^−^ at *m*/*z* 977.254 and MSE fragments at *m*/*z* 815,285], sinapate derivative, sinapoyl malate (compound 10, [(M−H)^−^ at *m*/*z* 339,0724 and MSE fragments at 309,223,193,164,149,133], isorhamnetin (compound 13 [(M−H)^−^ at *m*/*z* 315,05 and MSE fragments at 284,255,151.

The UPLC–QTOF/MS analysis of Nightshade leaves revealed the identification of hydroxycinnamic acid esters: neochlorogenic acid (compound 3, [(M–H)^−^ at *m*/*z* 353,0851 and MSE fragments at 191,179,135], chlorogenic acid (compound 4, [(M–H)^−^ at *m*/*z* 353,0868 and MSE fragments at 191,85], caffeoyl malic acid (compound 5, [(M−H)^−^ at *m*/*z* 295,0449 and MSE fragments at 179,133], dicaffeoylquinic acid (compounds 8 and 9[(M−H)^−^ at *m/z* 515,118, 515,1208 and MSE fragments at 353,191,179,135 and 353,191,179,173,135,99, rutin (compound 6, [(M−H)^−^ at *m/z* 609,1448 and MSE fragments at 300,271], kaempferol derivative, kaempferol-3-*O*-rutinoside (compound 7, [(M−H)^−^ at *m/z* 593,1519 and MSE fragments at 285,223,193,149].

The raw Chinese cabbage leaves contained low levels of kaempferol derivatives^.^ (Table 4), compared to the dried leaf samples which revealed a higher concentration. The presence of kaempferol derivatives are comparable with previous reports on *Brassica napus* L. var. oleifera L leaves [22]; sinapoylglucoside was also reported in a few Brassica species [23]. Conversely, kaempferol-sophoroside and ferulic acid were detected in a previous study but not in the samples harvested during summer (December to January) [3]. The summer temperatures and light radiation could have favoured the increased production of kaempferol, glycosylation with sugar molecules and sophorotrioses and acylation with ferulic acid [24]. Kaempferol-3-*O*-hydroxyferoyl-trihexoside was predominant among the four identified kaempferol derivatives in Chinese cabbage (Table 4). Although the dehydration treatments, such as microwave, solar, and freeze-drying treatments, enhanced the accumulation of kaempferol-sophoroside-hexoside, kaempferol-3-O-hydroxyferoyl-trihexoside, kaempferol-3-*O*-hydroxyferoyl-diglucoside, kaempferol-dihexoside and kaempferol-3-*O*-sinapoyldihexoside-hexoside in Chinese cabbage (Table 4), overall solar drying showed moderate accumulation of kaempferol derivatives. In a previous study, kaempferol glycoside was shown to increase from 316.08–444.69 to 340.16–486.51 µg g^−^^1^ in four cultivars of lentils during thermal processing [25]. The observed changes in kaempferol derivatives related to different drying methods suggests that the temperature changes could have affected the activity acyl-transferase responsible for acylation of kaempferol glycosides in varying degrees [26] and promoted the shift to the acylated kaempferol glycoside. Furthermore, glycosylation and acylation status of the flavonoids determines thermostability of flavonoids during thermal treatments [27] and acylated kaempferol tri- or tetra-glycosides are more thermally resistant during domestic food preparation [28]. 

Generally, a positive effect of heating on flavonoid glycosides has previously been shown in onion varieties heated up to 120 °C, and thereafter declined slightly at 150 °C for 30 min [29]. The activity of flavonoid glucosyltransferase responsible for the biotransformation of the flavonoid aglycones into *O*-glycosylated flavonoids was reported to be high at higher temperature conditions [30] and this could have facilitated multiple glycosylated products and extra di-glucosides, and tri-glucosides. The proportion of reduced diglucoside was reported in severely oven roasted onions [31]. The drying temperatures inside the oven was set at 100 °C in this study, which could have maintained the stability of diglucoside in Chinese cabbage leaves. In general, sun drying revealed a negative impact on the concentration of all four types of kaempferol derivatives in Chinese cabbage (Table 4).

It is interesting to note that kaempferol-3-*O*-rutinoside was in microwave dried Nightshade leaves were more or less similar to the raw leave samples and also showed the lowest levels when compared with the other dried samples, and the highest levels were detected in freeze-dried samples (Table 4). The freeze-drying method was mostly regarded as the best method of postharvest drying for the preservation of phenolic compounds because it changes the microstructure of the plant tissue so that the extraction of the flavonoids can be made easier [32]. However, freeze-dried Chinese cabbage and Nightshade leaves showed slight or moderate retention of flavonoid glycosides (isorhamantin glycosides, rutin) (Table 5 and Table 6). Conversely, the thermal processing increased the proportion of flavonoids than freeze-drying in the European cranberry (*Oxycoccus palustris* Pers.) [33]. The exposure of the Nightshade leaves for a shorter duration at a higher temperature during microwave drying, could have facilitated a faster rate of water loss and affected the flavonoid metabolism [34] and the metabolites (isorhamnetin-*O*-hexoside, caffeoylmalic acid, neochlorogenic and chlorogenic acids). 

Isorhamnetin-*O*-dihexoside was more prominent than the mono hexosides in the dried Chinese cabbage, but only isorhamnetin-*O*-hexoside was detected in the dried Nightshade leaves (Table 5). However, the concentrations of isorhamnetin-O-hexoside in the dried Nightshade leaves were lower than the concentrations noted in Chinese cabbage (Table 5). Isorhamnetin glycosides were not detected previously in our studies in Chinese cabbage raw leaf samples harvested during winter [3]. The concentration of isorhamnetin glycosides were shown to increase with the intensity of global radiation [24,35]. The Chinese cabbage leaves harvested during summer contained isorhamnetin glycosides, but the concentrations were much lower than the dried leaves. Raw Chinese cabbage leaves harvested in summer contained 2.34 mg kg^−^^1^ as stated in Table 5. Sun drying significantly reduced the isorhamnetin -*O*-dihexoside in Chinese cabbage (Table 5) probably due to the biotransformation to monhexoside or being destroyed due to direct sun light [34]; further investigations are needed to find the reasons for the observed changes. In contrast, solar dried Nightshade leaves showed the highest levels of isorhamnetin-*O*-hexoside (Table 6). The microwave and sun drying showed lower levers of isorhamnetin-*O*-hexoside (Table 6). This further demonstrates that the isorhamnetin-*O*-hexoside could have undergone transformation or decomposition with direct heating during microwave or sun drying. A similar increase in isorhamnetin glycoside (isorhamnetin-3-*O*-glucoside) by 56.98%, compared to the fresh black grapes (Ekşikara) (*Vitis vinifera* L.), was previously reported during sun drying [35]. Sinapoyl esters, sinapoyl malate, were reported in Pak choi (*Brassica rapa* subsp. chinensis) and it was reported that the concentrations depended on the developmental stage and radiation during production [36]. In this study, all drying treatments increased the concentration of sinapoyl malate compared to the fresh leaves of Chinese cabbage (Table 6), while the solar dried leaves revealed the highest concentrations. Pak choi breads, baked where crumb temperature was at 98 °C and the bread crust was set at 180 °C, showed a decline in sinapoyl malate [27,37]. However, in this study, the freeze drying process was shown to affect stability of sinapoyl malate (Table 6). Concentration of rutin was significantly higher in dried leaves samples than the raw Nightshade leaves (Table 6). Microwave drying significantly increased the levels of rutin in Nightshade leaves (Table 6). The temperature at 121 °C during hydrothermally treatment [38] and higher frying temperatures (150–170 °C) [39] did not severely reduce the concentration of rutin in buckwheat flour and the Tartary buckwheat instant fried noodle system. However, traditional sun drying affected the retention of rutin in Nightshade leaves (Table 6) and other factors, such as pH, affected the stability of flavonoids in a food matrix. Although the 3-hydroxy-function at the C-ring of flavonoid plays a major role during thermal degradation, the position of the sugar moiety was reported to block its degradation [40], in this study solar drying and sun drying was shown to affect the rutin concentration significantly. Rutin possess numerous health benefits, such as anti-diabetic activity, protection against neurodegenerative effects, and for the treatment of Alzheimer’s disease [41].

Caffeoylmalic, and neochlorogenic acids (isomer of chlorogenic acid) were detected in lower concentrations in raw Nightshade leaves [3] These two caffeic acid derivatives and chlorogenic acid (5-*O*-caffeicquinic acid) significantly increased in Nightshade leaves with different types of drying treatments, however, the oven drying method revealed a significantly higher concentration of caffeoylmalic acid (Table 6), while both solar and oven drying increased the concentration of chlorogenic and neochlorogenic acids significantly (Table 6). Amongst the three types of hydro cinnamic acid esters, the proportion of caffeoylmalic acid is higher in the dried Nightshade leaves (Table 6). In contrast, during the roasting of coffee beans, the concentration of chlorogenic acid decreased [41]. The observed discrepancies between our observation and Król et al., [42] can be attributed to the temperature and duration of the adopted thermal treatments. The higher temperatures were reported to facilitate the non-enzymatic degradation of chlorogenic acid [42]. Chlorogenic acid shows anti-diabetic, anti-obesity effects and benefits weight loss [42]. Microwave drying treatments negatively affected the concentration of caffeoylmalic, chlorogenic and neochlorogenic acids (Table 6). On the contrary, freeze-drying was shown to increase the chlorogenic acid content in stevia leaf [43], and apricots cv. Cafona dried at 55 °C showed similar reduced retention of neochlorogenic and chlorogenic acids in freeze-dried samples most probably due to the activity of browning enzyme PPO [44]. High-temperature thermal treatment was shown to increase caffeic acid derivatives [45]. Further investigation on the transformation of 5-*O*-caffeoylquinic acid to other 3- and 4-*O*-caffeoylquinic acids was not performed in this study due to lack of analytical standards, thus it can be suggested that the accumulation of phenolic compounds was regulated due to the drying condition related to temperature with a characteristic pattern for each compound. Chlorogenic acid has a profound impact on health due to several biological activities, such as anti-inflammatory and anti-diabetic activity related to type 2 diabetes due to the inhibition of α-amylase and α-glucosidase activities, and anti-obesity properties [46]. In addition, caffeoylmalic acid was also reported to show anti-inflammatory activity [45]. 

The differences between the phenolic metabolic profiles of the Chinese cabbage and Nightshade leaves using different drying methods was illustrated using an unsupervised Principal Component Analysis (PCA) approach utilising the data generated by the UPLC–281 Q-TOF/MS analysis. Both PC 1 and PC 2 for drying treatments of Chinese cabbage and Nightshade leaves explained more than 60% of the variance and revealed a good statistical separation among the different postharvest drying treatments (Figure 1A,B). The PCA plot, demonstrated two district groups based on the metabolites, and showed that drying treatments manipulated the distribution of metabolites in Chinese cabbage and Nightshade leaves (Figure 1A,B). Group 1 in Chinese cabbage included the traditional sun drying, and group 2 contained all the other drying methods, such as solar, oven, microwave and freeze-drying treatments. The group 1 in Nightshade contained the microwave dried leaves and the group 2 included the leaves subjected to the other drying methods.

These separation of sundried Chinese cabbage and the other treatments are supported quantitatively based on the phenolic metabolites in Table 4, Table 5 and Table 6. The clear separation of the two groups of sundried Chinese cabbage from other drying treatment s reveal that most of the phenolic metabolites are present at a lower concentrations in sun dried Chinese cabbage. Similarly, majority of the phenolic metabolites were lower in concentration in dried Nightshade leaves compared to those leaves subjected to the other types of drying treatments (Table 4 and Table 6).

Furthermore, supervised Orthogonal Projections to Latent Structures Discriminant Analysis (OPLS-DA) was executed to identify the marker candidates responsible for better discrimination. 

The values of R^2^ X and Q^2^ of cross-validation in OPLS-DA score plot for Chinese cabbage were 0.983 and 0.946, respectively. For Nightshade leaves, values of R^2^ X and Q^2^ of cross-validation in OPLS-DA score plot were 0.910 and 88.76 and both the models revealed excellent fitness and reliability. Furthermore, S-plot were plotted to identify the key metabolites that were responsible for the separation of sundried Chinese cabbage from the other types of drying treatments and similarly the microwaved Nightshade and the other types of drying treatments. 

Based on the S-plot, the most important candidate markers for the observed separation for the drying treatments of Chinese cabbage are given in Table 7.

Kaempferol-3-O-sophoroside (601.3231), kaempferol-3-sophorotrioside-7-glucoside (933.2275) and hydroxyoctadecenedioic acid (327.2171) were the markers responsible for the separation of sun-dried samples from the other drying treatments in Chinese cabbage (Table 7). Whereas, sinapoyl malate (339.0712) and the unidentified compound (325.0555) acted as candidate markers for the separation of samples obtained from the other drying treatments from the sundried samples. 

Although S-plot showed five marker compounds as shown in Table 7, the Figure 2 illustrates the quantitative differences of the marker compounds at Rt 3.46, 3.98, 9.78, and 2.95, that revealed abundance of the following unidentified compound [M−H] 325, sinapoyl malate (339), kaempferol-3-*O*-sophoroside (601.3), kaempferol-3-sophorotrioside-7-glucoside (933.22) at 199, 800, 100, and 400 peak intensity in counts sec^−1^ respectively in traditional sun dried Chinese cabbage leaf samples, whilst in samples from other drying treatments (traditional sun, solar, oven and freeze-drying), revealed abundance of unidentified compound, sinapoyl malate, kaempferol-3-*O*-sophoroside and kaempferol-3-sophorotrioside-7-glucoside at Rt 3.46, 9.78, and 2.95 showing [M−H] 400, 1200, 0, and 200 peak intensity in counts sec ^−1^ respectively. However, kaempferol-3-*O*-sophoroside] 601.3 was not detected. 

For Nightshade, the S-plot, revealed hydroxyoctadecanedioic acid (329.2323), trihydroxyoctadecadiene derivatives (327.2171, 327.2166) separated the microwaved samples from the samples that underwent other types of drying treatments (Table 8). Thus, these compounds can be regarded as the candidate markers for the microwaved samples. Whereas, caffeoylmalic acid (591.0983) was detected in the samples that underwent other types of drying treatments and this compound as can be considered as a biomarker (Table 8). The Figure 3 illustrates the quantitative difference of these marker compounds at Rt 7.46, 6.36, 6.83, and 3.37 showing the abundance of hydroxyoctadecanedioic acid (329.2323) at 50 peak intensity in counts sec^−1,^ trihydroxyoctadecadiene derivatives (327.2171, 327.2166) at 300 and 100, caffeoyl malic acid (591.0983) 950,peak intensity in counts sec^−1^ respectively in Nightshade leaf samples underwent in other drying treatments (traditional sun, solar, oven and freeze-drying), whilst in microwave dried samples at Rt 7.46, 6.36, 9.78 and 3.37, the abundance of hydroxyoctadecanedioic acid, was at 700 peak intensity in counts sec^−1^, trihydroxyoctadecadiene derivatives were at 1200 and 1400, peak intensity in counts sec^−1^ and caffeoylmalic acid, revealed 25 peak intensity in counts sec ^−1^.

### 2.3. Saponins 

Saponins are responsible for the bitter taste, but possess antidiabetic, anti-obesity and anticarcinogenic activity [47]. Anti-nutritive compound saponins were tentatively identified in Nightshade leaves based on the UPLC–QTOF/MS analysis and previous research reports [46,47]. The compound 10 (*m*/*z* 1227,5996), compound 11 (*m/z* 1213,585) and compound 12 (*m/z* 1313,6348) were tentatively identified as tigogenin-G-G-G-G-G (5 glucose units attached) Tigogenin-G-G-G-Xyl-G (four glucose units and one xylose) and Tigogenin-G-G-Rha_Xyl-Xyl (2 glucose + rhamnose+ 2 xylose units) respectively (Table 3b, Appendix A). 

Tigogenin-5G and Tigogenin-G-G-Rha_Xyl-Xyl were observed to increase during drying treatments at significantly lower concentrations in raw leaf samples (Table 9); both tigogenin compounds reached the highest concentration during freeze-drying. Tigogenin-G-G-Rha_Xyl-Xyl was not detected in microwaved Nightshade leaves. Microwave drying demonstrated the lowest concentration of saponin compounds compared to the other drying treatments. Daily consumption of Nightshade vegetables in large quantities must be limited due to the toxic properties of saponins causing haemolysis of red blood cells [48].

### 2.4. Ascorbic Acid Content 

The ascorbic acid content of the fresh and freeze-dried Chinese cabbage and Nightshade leaves did not differ significantly (Table 9). Similar observation was reported in five tropical fruits, namely starfruit (*Averrhoa carambola* L.), mango (*Mangifera indica* L.), papaya (*Carica papaya* L.), muskmelon (*Cucumis melo* L.), and watermelon *Citruluss lanatus* (Thunb.) [49]. Freeze dried Chinese cabbage and Nightshade leaves samples demonstrated higher concentration of ascorbic acid compared to the oven and microwave dried samples (Table 10). Similar observation was noted in freeze dried Moringa leaves [50]. In comparison, heat dried moringa leaves, especially oven and microwave dried samples, showed higher antioxidant activity (DPPH assay) [51]. Ascorbic acid (vitamin C) was shown to degrade during thermal processing and microwave heating revealed the loss of ascorbic acid in some cases more than the conventional drying treatments [51]; this was probably due to the light energy during microwave drying. However, in this study conventional higher temperature (100 °C) during oven drying revealed the lowest concentration of ascorbic acid in both vegetables (Table 10). 

### 2.5. Antioxidant Property 

The antioxidant capacity (FRAP assay) and the scavenging activities (DPPH assay) were evaluated for dried Chinese cabbage and Nightshade leaves, and are presented in Table 11 and Table 12. Traditional sun drying was shown to have a negative impact on Chinese cabbage leaves, in Nightshade leaves, the microwave drying was noted to reduce the antioxidant activity, while the impact of freeze-drying on the antioxidant properties was positively expressed in both types of leafy vegetables (Table 11 and Table 12). Antioxidant property of the vegetables depends on the presence of antioxidant constituents and the biosynthesis of novel antioxidants during postharvest processing. The thermal drying process was expected to increase the antioxidant property of both vegetables. However, heating at 65 °C or 100 °C negatively affected the antioxidant activities in food products [52]. The non-thermal freeze-drying process could have preserved the other non-phenolic antioxidants, such as ascorbic acid and carotenoids, that have higher redox potential and contributes more towards the DPPH and ABTS^+^ activities [53]. Moreover, the presence of a number of hydroxyl groups of phenolic compounds determined the antioxidant activity [53]. In glycosylated flavonoids, the glycosylation presence of mono or diglycosidic molecules and the arrangement of hydroxyl group on the flavonoid B and C rings affects the antioxidant property [53]. 

Thus, this finding suggests that freeze-drying of Chinese cabbage and Nightshade leaves are preferred to the other drying methods to preserve the antioxidant property of both vegetables.

### 2.6. Proximate Analysis of Solar Dried Functional Powder

The proximate analysis of solar dried Chinese cabbage and Nightshade functional powders are given in Table 13. Total sugar, carbohydrate and sodium concentrations were higher in solar dried Nightshade leaf powder compared to Chinese cabbage (Table 13). Low sodium content makes it acceptable for people with high blood pressure and kidney problems. At the same time, sodium is a vital intracellular and extracellular cation and assists in the regulation of plasma volume and acid-based balance during nerve and muscle contraction [54]. However, the fat content was higher than the solar dried Nightshade and lower than the Chinese cabbage (Table 13).

Total dietary fibre content, protein, ash, carbohydrate, fat, and the energy levels in both Chinese cabbage and Nightshade solar dried functional powders were higher than the levels reported in solar dried Yanrin (*Laurea Taraxifoli*) [55]. A lower carbohydrate content is associated with a lower calorific content of food; 100 gm Nightshade and Chinese cabbage leaves contained calorific content of (energy) 1148.13 kJ 100g^−1^ and 1118.67 kJ 100g^−1^ respectively. It is also interesting to note that the solar dried cowpea leafy vegetable (*Vigna unguiculata* L.) contains lower protein (29.40 g), carbohydrate (23.5 g) fibre (17.10 g) and ash (10.60 g). Higher values of ash content relate to the higher mineral composition [56]. The solar dried samples in this study showed lower moisture content than the solar dried cowpea leafy vegetable. Lower moisture or water activity is important to protect the functional powders from microbial infestation leading to physico-chemical changes and the shelf of the product [57]. Moisture content of the product also depends on the efficiency of the solar dryers used. Solar dried pumpkin (*Telferia occidentalis*) leaves contained higher carbohydrate content (66.6% to 67.0%) than the solar dried Chinese cabbage and Nightshade leaves [58]. It is noteworthy to mention that the protein, ash and dietary fibre in solar dried Nightshade and Chinese cabbage leaves were much higher than the solar dried moringa (*Moringa oleifera*) leaves [59]. Dietary fibre consists of digestion-resistant cellulose, hemicellulose, pectic substances, gums, mucilage and lignin and is proposedly important for a healthy gut microbiome and reported to lower glycaemic indices of food [60]. Solar dried moringa contained 27.84%–28.36% protein, 1.03%–1.06% ash, 7.32%–7.26% fibre, but the carbohydrate content (48.64%–50.27%) [59] was higher in solar dried moringa, than the solar dried Nightshade and Chinese cabbage leaves. Thus, it can be concluded that the solar dried Nightshade and Chinese cabbage can be regarded as source of protein. 

## 3. Materials and Methods 

### 3.1. Raw Materials

Nightshade (*Solanum retroflexum* Dun.) and Chinese cabbage (*Brassica rapa* L. subsp. *chinensis*) leaves were obtained from Tshiombo irrigation scheme in Venda, Limpopo, South Africa. The leaves were harvested in the early spring of 2018. The leaves at 8-leaf stage, reached after 60 to 95 days of planting [58], and free from dirt, pest damage or decay were selectively harvested by being detached from the stem and washed in tap water. 

### 3.2. Chemicals 

The standards of methanol, hexane, acetone, iron chloride, metaphosphoric acid, 2,6-dichlorophenol-indophenol dye, sodium chloride, sodium carbonate, sodium nitrite, aluminium chloride, hydrochloric acid (37%), sodium hydroxide, sodium acetate trihydrate, glacial acetic acid, 2,4,6-tripyridyl-s-triazine (TPTZ), 2,2′-azobis(2- amidinopropane) hydrochloride, ferric chloride, potassium persulfate, monosodium phosphate, disodium phosphate, sodium acetate, vanillin, hydrochloric acid, sulfuric acid, nitric acid, methanol, tannic acid, monohydrate ferric chloride, trolox, sulfosalicylic acid, oxalate, calcium chloride, ammonium hydroxide, potassium permanganate analytical standard chlorogenic acid ≥ 95%), catechin (≥ 95%), luteolin (≥ 95%), epicatechin (≥ 95%) and rutin (≥ 95%) and gallic acid (≥ 95%) were purchased from Sigma Aldrich, Johannesburg, South Africa.

### 3.3. Postharvest Drying Treatments

Dying methods were carried out using a solar dryer hot air oven, microwave oven and freeze dryer. Solar drying was carried out in a solar dryer erected for the Tshiombo tribal council community jointly with Scaled Impact Pty (Ltd), Johannesburg and Tshwane University of Technology, Pretoria, South Africa. Uniform size leaves (380 g) were placed in 20 trays and the uniform warm air was circulated through the trays ((894 × 605 × 80 mm) within the cabinet with the help of fans mechanised with a solar photovoltaic panel. The average temperature within the solar cabinet was set at 45 to 50 °C, and the RH to around 30% to 45% based on the weather conditions. On a warm day, it took 48 h to dry the Chinese cabbage and Nightshade leaves separately until constant weight was achieved. 

Oven drying was performed by placing the samples of Nightshade leaves separately (5–6 kg per tray) on a stainless-steel tray (915 × 840 × 915 mm) and using a drying oven (95 L) with Forced Convection System (Digital series, EcoTherm, Hartkirchen, Austria) at 100 °C, RH 12% to 14% until constant weight (2 days).

For freeze-drying the leaf samples were frozen at −80 °C and thereafter freeze-dried using a Vacuum freeze-drier (YK-118-50, Taiwan) at −47 °C to −53 °C for 72 h. 

Microwave drying was carried out in a domestic digital microwave oven (Samsung 40l microwave oven, Seoul, South Korea) with technological properties of 230 V, 50 Hz, and frequency of 2450 MHz (a wavelength of 12.24 cm) and output powers at 900 W for 15 min (the output power and the time were selected based on preliminary trials) using 25 g of leaf sample placed and spread evenly on the rotating glass platform of the microwave. Digital control of the microwave oven was used to regulate the output power and processing time. The traditionally adopted “open sun-drying” was performed by spreading the leaves (∼100 g) on wooden trays for 8 to 12 h at 25 °C. Moisture loss was monitored repeatedly for each drying process until the moisture content was between 7% and 9%.

### 3.4. Colour Changes 

The dried Chinese cabbage and Nightshade leaves were ground separately, in order to produce a homogeneous powder, with a domestic coffee grinder for 30 s. The colour analysis was performed in comparison with the un-ground fresh leaves. Colour measurement was taken with a Colour meter (CM-700d, Konica Minolta Sensing Inc, Tokyo, Japan), according the method previously described by Managa et al., [3]. The colour coordinates *L***, a***, b** were recorded to calculate the total colour change ∆ *E* to determine the colour difference between fresh and dried leaves. Total colour change was calculated to evaluate the effect of the treatments on colour retention. Fresh leaf samples’ colour coordinate values are *L*_1_***, *a*_1_*, *b*_1_*** and the samples from different drying treatments represent *L*_2_*, *a*_2_*, and *b*_2_* [3].

### 3.5. Lipophilic Pigments 

The chlorophyll’s a (*Chl a*) and b (*Chl b*), total chlorophyll and carotenoid content were determined following the method described by Mampholo et al., [1] and Managa et al., [3]. The chlorophyll a (Chl a) b (Chl b), and total chlorophyll were quantified without modifications as previously described by Managa et al., [3], using leaf samples (0.2 g) ground with 2 mL acetone and hexane 4:6 (*v/v*) and extracting for 2 h. Thereafter, the sample mixture was centrifuged for 10 min at 4◦C (9558×g). Afterwards, the supernatant was decanted, and a portion of the solution was measured at 470, 646 and 662 nm (Biochrom Anthos Zenyth 200 Microplate Reader; SMM Instruments, Biochrom Ltd., Johannesburg, South Africa). The Chl a and Chl b contents were determined according to equations: Chl a=15.65A662 − 7.340A646, Chl b=27.05A646 −11.21A662. The content of Chl a + Chl b gives the total chlorophyll content. Total carotenoids = (1000A470 − 2.270Chl a − 81.4Chlb)/227. The total chlorophyll and carotenoids were expressed in mg per 100g in fresh weight basis

### 3.6. Untargeted Phenolic Metabolites

Identification and quantification of predominant polyphenolic metabolites were performed using the Quadrupole time-of-flight (QTOF) mass spectrometer (MS) UPLC-QTOF/MS (Waters, Milford, MA, USA, as described earlier by Managa et al. [3] and Ndou et al. [61] using 5 g sample of snap frozen Chinese cabbage and Nightshade leaves from oven drying, solar drying, microwave drying and freeze drying. The conditions for separation of the phenolic compounds are Ndou et al. [58]. The identification and quantification of the phenolic components were carried out using cocktail standards chlorogenic acid, catechin, luteolin, epicatechin and rutin, as previously described [3,58], due to the to the unavailability of the calibration standards for all the compounds. The obtained data was processed using the established TargetLynx method to produce integrated peak areas for each compound as described in detail previously [3,59].

### 3.7. Trolox Equivalent Antioxidant Capacity (TEAC) ABTS, and FRAP

For the detection of Trolox equivalent antioxidant capacity (TEAC) the 2.5 mM 2,2′-azobis(2- amidinopropane) hydrochloride (ABAP) and 20 mM 2,2′- azinobis(3-ethylbenzothiazoline-6-sulfonate) ABTS 2 stock solution in 100 mL of phosphate buffer (100 mM phosphate and 150 mM NaCl, pH 7.4) were mixed and incubated at 60 °C for 6 min without any modifications as described by Managa et al. [3] and Egea, Sánchez-Bel, Romojaro, Pretel [62] to produce the ABTS ^-^ radical anion. Afterwards the mixture was held in darkness for 16 h at 25 °C and diluted with 0.1 mM phosphate buffer (pH 7.0) until to obtain an absorbance at 734 nm (1.1 ± 0.002 units). Thereafter, the radical solution (285 μL) was added to the sample extract (15 μL) and the decrease in absorbance observed at 734 nm for 6 min was used to calculate the Trolox equivalent antioxidant capacity (TEAC). Calibration curves were constructed for each assay using different concentrations (0−20 mg) of Trolox. The antioxidant activity (ABTS assay) was expressed as µmg of TEAC g FW^−^^1^. 

Ferric reducing antioxidant power assay was executed following method described by Mpai et al. [63]. Nightshade leaf samples (0.2 g) were homogenized in 2 mL sodium acetate buffer at pH of 3.6. The ferric reducing ability was estimated by mixing 15 μL aliquot of leaf extract, with 220 μL of FRAP reagent solution [10 mmol L^−^^1^ TPTZ [2,4,6-tris(2-pyridyl)-1,3,5-triazine] acidified with concentrated HCl, and 20 mmol L^−1^ FeCl_3_]. The absorbance was read at 593 nm and the reducing antioxidant power content was calculated using a standard curve of Trolox and expressed μmol TEAC g^−1^ FW. 

### 3.8. Ascorbic Acid Content 

Ascorbic acid content was determined from 30 g leaf samples five replicates from each drying treatment, according to Dinesh et al., [64], using the 2,6-dichlorophenol-indophenol titration method. DW. Five grams were 10 mL of 4% oxalic acid and the mixture was ground and filtered and the final volume was made 25 mL with 4% oxalic acid. Afterwards, 5 mL of ascorbic acid standard (500 µg 5 mL^−1^) and 10 mL of 4% oxalic acid were pipetted into a 100 mL conical flask and the contents in the flask were titrated against the 2,6-dichlorophenol-indophenol until the appearance of a pale pink colour that persisted for a few min. Thereafter, 5 mL of the sample solution was titrated in the same manner, against the 2,6-dichlorophenol-indophenol solution. Ascorbic acid content present in the test samples were determined using the formulae stated by Dinesh et al., [64], and expressed in mg 100 g^−1^

### 3.9. Proximate Analysis of Solar Dried Chinese Cabbage and Night Shade

Proximate analysis was performed on a selected fruit leather using standards methods using AOAC [65]. Leaf samples (100 g) were used for proximate analysis and five replicate samples were used. Kjeldahl method was adopted to quantify the nitrogen content and the nitrogen content was converted to protein by multiplying a factor of 6.25. The fat content was determined using hexane and soxhlet extraction method using 5 g of leaves samples. Fibre was quantified using 2 g of dried ground sample digested with 0.25 M H_2_SO_4_ and 0.3 M sodium hydroxide solution. Resulting insoluble residue was washed with warm water and dried in a hot air oven (LBOTEC, South Africa), at 100 °C until constant weight was obtained for the determination of fibre content. Ash content was determined by the incineration of a dried powdered sample (5 g) in a muffle furnace (Zhengzhou Protech Technology Co, China) at 550 °C for 12 h until the ash turned white. Carbohydrates were calculated according to the following formula: 100 – (% moisture + % proteins + % lipids + % ash +% fibres). Calorific value: (% proteins × 2.44) + (% carbohydrates × 3.57) + (%lipids × 8.37). Dried leaves (20 mg) were hydrolyzed with 3 ml of H_2_SO_4_ for 3 h at 100 °C. For the quantification of Na mineral, a set of 20 mg sample was mixed with 12 mL of HNO_3_ and incubated over night at room temperature. Thereafter, 4.0 mL perchloric acid (HCLO_4_) was added to this mixture and placed on the fume block. The temperature increased gradually and reached up to 250–300 °C and the digestion was completed in 70–85 min with the emergence of white fumes. After cooling down the temperature the contents were transferred to the glass tubes and 100 mL distilled water was added. This wet digested solution was used for the determination of Na content using the flame photometer and the calculations were done according to the AOAC [65].

### 3.10. Statistical Analysis 

A completely randomised design was adopted with 10 replicates per treatment and the experiments repeated twice. One-way analysis of variance (ANOVA) was used to analyse the significant differences between different postharvest drying treatments on different parameters at *p* < 0.05. Treatment means were separated using Fisher’s protected *t*-test least significant difference (LSD) at the 5% level of significance. Data were analysed using the statistical program GenStat for Windows (2004).

As shown previously by Managa et al., [3], the aligned data were imported using the PCA-X model with pareto scaling and analysed as an unsupervised multivariate cluster technique. Orthogonal projection to latent structure-discriminant analysis (OPLS-DA) was performed for classification and discriminant analysis of the sample. The loading plots were used to explain the association among the measured variables as previously [3], and those furthest from the origin of the plot were regarded as the highest contributors to variance in the chemical space. Approximately 20 samples per treatment were subjected to PCA and OPLS analysis. 

## 4. Conclusions

In conclusion, postharvest processing for functional food production requires the highest retention of beneficial phytochemicals. The different postharvest drying methods affected the phenolic metabolite composition on Chinese cabbage and Nightshade due to structural modifications. Although the antioxidant activity of the vegetables was maintained during freeze-drying, it is expensive to be implemented in the sub-Saharan rural regions, thus solar drying, which is regarded as a more hygienic way to dry leafy vegetables with shorter drying time, can be recommended to the farmers due to the moderate retention of antioxidant activity, chlorophyll carotenoid retention, protein, dietary fibre and reduced colour difference (*∆E*). 

## Figures and Tables

**Figure 1 molecules-25-01326-f001:**
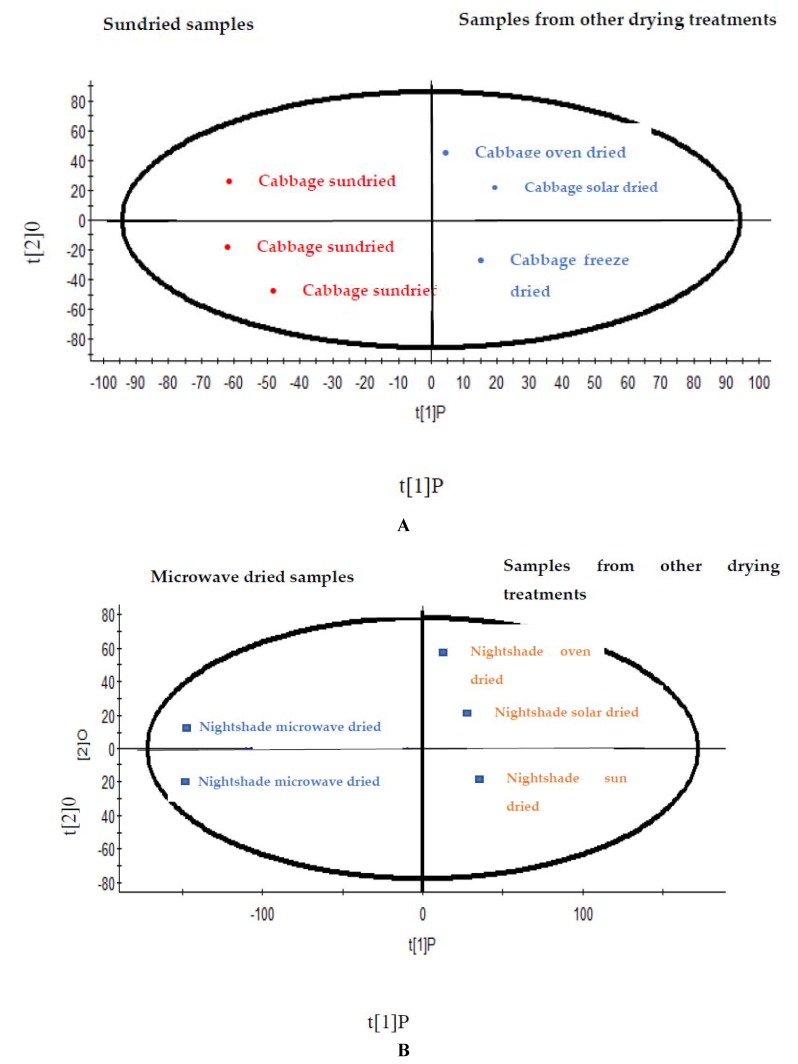
(**A**) Score plot of Principal component analysis (unsupervised) based on UPLC–Q-TOF/MS spectra of different drying treatments Chinese cabbage leaves. (**B**) Score plot of Principal component analysis (unsupervised) based on UPLC–Q-TOF/MS spectra of different drying treatments of Nightshade leaves.

**Figure 2 molecules-25-01326-f002:**
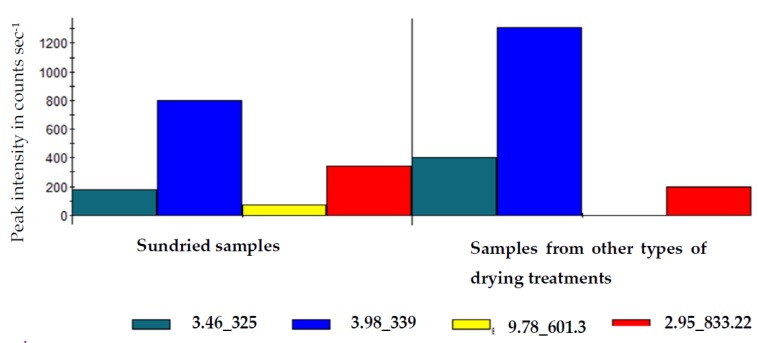
Histogram illustrating quantitative differentiation of biomarkers, between traditional sun drying and other drying treatments (traditional sun, solar, oven, and freeze-drying) of Chinese cabbage leaves.

**Figure 3 molecules-25-01326-f003:**
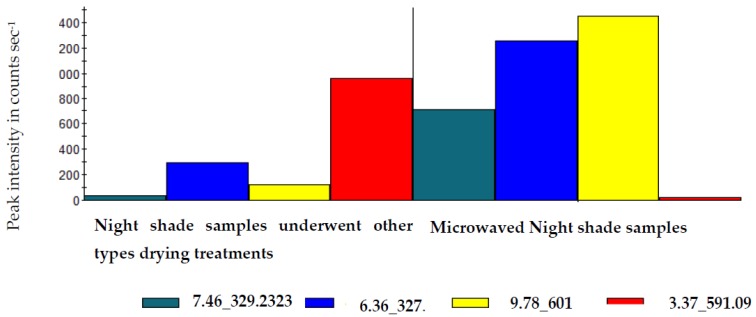
Histogram illustrating quantitative differentiation of biomarkers, between microwave drying and other drying treatments (traditional sun, solar, oven and freeze-drying) of Nightshade leaves.

**Table 1 molecules-25-01326-t001:** Effect of drying treatments on colour changes in Chinese cabbage and Nightshade leaves.

**(A) Effect of Drying Treatments on Colour Changes in Chinese Cabbage**					
**Treatments**	*L **	*a **	*b **	*h*°	*∆E*
Raw leaves	38.05 ± 0.0 ^d^	−7.74 ± 0.03 ^c^	13.21 ± 0.00 ^c^	119.09 ± 0.06 ^d^	0.00 ± 0.00 ^f^
Solar drying	48.96 ± 0.03 ^b^	−10.10 ± 0.06 ^e^	18.19± 0.10 ^b^	121.37 ± 0.01 ^b^	5.03 ± 0.03 ^d^
Microwave drying	36.93 ± 0.0 ^e^	−7.06 ± 0.01 ^b^	12.98± 0.05 ^d,c^	113.42 ± 0.01 ^e^	14.87 ± 0.05 ^b^
Freeze-drying	55.96 ± 0.11 ^a^	−16.04 ± 0.03 ^f^	26.54± 0.34 ^a^	125.99 ± 0.0.0 ^a^	2.79 ± 0.02 ^e^
Oven drying	35.01 ± 0.04 ^f^	−5.03 ± 0.02 ^a^	12.56± 0.01 ^d^	110.54 ± 0.01 ^f^	25.93 ± 0.21 ^a^
Sun drying	44.20 ± 0.04 ^c^	−9.04 ± 0.07 ^d^	17.97± 0.05 ^b^	119.38 ± 0.01 ^c^	10.62 ± 0.15 ^c^
**(B) Effect of Drying Treatments on Colour Changes in Nightshade Leaves**					
**Treatments**	*L **	*a **	*b **	*h°*	*∆E*
Raw leaves	42.89 ± 0.10 ^d^	−9.89 ± 0.03 ^d^	14.92 ± 0.01 ^d^	117.99 ± 0.05 ^d^	0.00 ± 0.00 ^f^
Solar drying	45.03 ± 0.01 ^b^	−7.03 ± 0.01 ^c^	16.02 ± 0.08 ^c^	122.10 ± 0.11 ^b^	3.58 ± 0.06 ^d^
Microwave drying	35.13 ± 0.00 ^e^	−4.43 ± 0.01 ^b^	9.01 ± 0.08 ^e^	114.04 ± 0.08 ^e^	13.45 ± 0.03 ^b^
Freeze-drying	47.08 ± 0.01 ^a^	−12.35 ± 0.00 ^e^	19.78 ± 0.01 ^a^	124.53 ± 0.25 ^a^	3.18 ± 0.02 ^e^
Oven drying	33.11 ± 0.07 ^f^	−2.98 ± 0.00 ^a^	8.36 ± 0.00 ^f^	108.05 ± 0.06 ^f^	14.66 ± 0.05 ^a^
Sun drying	44.38 ± 0.34 ^c^	−9.99 ± 0.08 ^d^	17.77 ± 0.01 ^b^	120.04 ± 0.07 ^c^	5.81 ± 0.01 ^c^

Means followed by the same letter within the column are not significantly different, **p* < 0.05 level.

**Table 2 molecules-25-01326-t002:** Effect of drying treatments on lipophilic changes in Chinese cabbage and Nightshade leaves.

**(A) Effect of Drying Treatments on Lipophilic Changes in Chinese Cabbage Leaves**				
Treatments	Chlorophyll a *(Chl a)*	Chlorophyll b *(Chl b)*	Total chlorophyll (*Chl*)	Carotenoids
		(mg 100 g^−1^)		
Raw leaves	3.73 ± 0.06 ^d^	2.75 ± 0.03 ^d^	6.30 ± 0.02 ^d^	0.93 ± 0.00 ^d^
Solar drying	4.33 ± 0.11 ^b^	3.48 ± 0.06 ^b^	7.29 ± 0.03 ^b^	1.75 ± 0.00 ^b^
Microwave drying	4.50 ± 0.03 ^b^	2.63 ± 0.03 ^e^	6.95 ± 0.01 ^c^	0.67 ± 0.00 ^d^
Freeze-drying	13.58 ± 0.08 ^a^	6.03 ± 0.08 ^a^	19.62 ± 0.03 ^a^	1.81 ± 0.00 ^a^
Oven drying	3.67 ± 0.01 ^c^	2.46 ± 0.01 ^e^	7.08 ± 0.00 ^c^	0.55 ± 0.01 ^e^
Sun drying	4.29 ± 0.03 ^b^	2.99 ± 0.11 ^c^	7.13 ± 0.02 ^c^	1.01 ± 0.02 ^c^
**(B) Effect of Drying Treatments on Lipophilic Changes in Nightshade Leaves**				
Treatments	Chlorophyll a *(Chl a)*	Chlorophyll b *(Chl b)*	Total chlorophyll (*Chl*)	Carotenoids
		(mg 100g^−1^)		
Raw leaves	3.73 ± 0.06 ^d^	2.75 ± 0.03 ^d^	6.30 ± 0.02 ^d^	0.93 ± 0.00 ^d^
Solar drying	4.33 ± 0.11 ^b^	3.48 ± 0.06 ^b^	7.29 ± 0.03 ^b^	1.75 ± 0.00 ^b^
Microwave drying	4.50 ± 0.03 ^b^	2.63 ± 0.03 ^e^	6.95 ± 0.01 ^c^	0.67 ± 0.00 ^d^
Freeze-drying	13.58 ± 0.08 ^a^	6.03 ± 0.08 ^a^	19.62 ± 0.03 ^a^	1.81 ± 0.00 ^a^
Oven drying	3.67 ± 0.01 ^c^	2.46 ± 0.01 ^e^	7.08 ± 0.00 ^c^	0.55 ± 0.01 ^e^
Sun drying	4.29 ± 0.03 ^b^	2.99 ± 0.11 ^c^	7.13 ± 0.02 ^c^	1.01 ± 0.02 ^c^

Means followed by the same letter within the column are not significantly different, *p* < 0.05 level.

**Table 3 molecules-25-01326-t003:** Tentative peak assignment of the metabolites contained in Chinese cabbage and Nightshade leaves exposed to different postharvest drying methods.

**(A) Tentative Peak Assignment of the Metabolites Contained in Chinese Cabbage Leaves Exposed to Different Postharvest Drying Methods**							
**No**	Retention time	[M-H]^−^	M-H formula	ppm error	MSE fragments	UV	Tentative identification
**1**	1.12	191.05	C6H7O7	1.6	111,97,87	weak	Citric acid
**2**	2.67	203.03	C11H11N2O2	1.0	116,74	270	Tryptophan
**3**	2.71	771.21	C33H39O21	3.5	609,285,255	265,347	Kaempferol-3-*O*-sophoroside 7-Oglucoside
**4**	2.85	963.2401	C43H47O25	−0.5	801,609,285	267,333	Kaempferol-3-*O*-hydroxyferuloyl-trihexoside
**5**	3.9	609.1474	C27H39O16	3.0	447,285	265,321	Kaempferol-di-hexoside
**6**	3.12	977.254	C44H49O25	−0.3	815,285	265,333	Kaempferol-3-*O*-sinapoyldihexoside-hexoside
**7**	3.17	639.1542	C28H31O17	−1.6	477,315	254,340	Isorhamnetin-*O*-dihexoside
**8**	3.45	431.1922	C20H31O10	1.2	325,209,207,163,150,133	327	Malic acid derivative
**9**	3.59	801.1874	C37H37O20	−0.5	771,609,285,255	265,340	Kaempferol-3-*O*-hydroxyferuloyl dihexoside
**10**	3.96	339.0724	C15H15O9	2.4	309,223,193,164,149,133	265,340	Sinapoyl malate
**11**	4.07	447.1033	C22H21O12	0.0	314	253,350	Isorhamnetin-*O*-hexoside
**12**	4.20	315.05	C16H11O7	−1.6	284,255,151	291	Rhamnetin/Isorhamnetin
**13**	4.63	363.0714	C17H16O9	−0.6	345,285,271	289	Kaempferol-dimethoxy derivative Pubchem 123947790
**14**	6.35	327.2163	C18H31O5	−2.1	229,211,171,97	weak	Hydroxyoctadecenedioic acid.
**15**	6.82	329.2327	C18H33O5	1.2	223,211,171,139,99	275	Hydroxyoctadecanedioic acid
**16**	9.10	531.2821	C26H43O11	0.6	485,307	none	Cinncassiol-glucoside
**(B) Tentative Peak Assignment of the Metabolites Contained in Nightshade Leaves Exposed to Different Postharvest Drying Methods**							
**No**	Retention time	[M-H]^−^	M-H formula	ppm error	MSE fragments	UV	Tentative identification
**1**	0.80	133.0145	C4H5O5	−1.5	115,96,71	266	Malic acid
**2**	0.92	191.0195	C6H7O7	1.6	111,97,87	weak	Citric acid
**3**	2.66	353.0851	C16H17O9	−2.5	191,179,135	325	Neochlorogenic acid
**4**	2.98	353.0868	C16H17O9	−2.0	191,85	325	Chlorogenic acid
**5**	3.40	295.0449	C13H11O8	−2.7	179,133	328	Caffeoyl malic acid
**6**	3.73	609.1448	C27H29O16	1.6	300,271	255,352	Rutin
**7**	3.90	593.1519	C28H29O15	2.4	285,223,193,149	331	Kaempferol 3-O-rutinoside
**8**	4.12	515.118	C25H23O12	−1.9	353,191,179,135	327	Dicaffeoylquinic acid
**9**	4.31	515.1208	C25H23O12	3.5	353,191,179,173,135,99	327	Dicaffeoylquinic acid
**10**	5.08	1243.598	C57H94O28	1.9	1081,919,757	none	Tigogenin_5G
**11**	5.16	1213.585	C79H93O18	6.8	1081,1051,919,757	none	Tigogenin-3G-Xyl-G
**12**	5.60	1329.631	C68H97O26	−0.7	1081, 919	none	Tigogenin-GG-Rha-Xyl_Xyl
**13**	6.37	327.2165	C18H31O5	−2.1	229,211,171,97	310	Trihydroxyoctadecadiene derivative
**14**	6.91	329.2328	C18H33O5	−3.0	201,171,99	300	Hydroxyoctadecanedioic acid
**15**	8.76	309.2054	C18H29O4	−4.2	220,187,97	none	Undecylresorcylic acid
**16**	9.07	307.1908	C18H27O4	−0.3	222,121,99	280	Hydroxy-oxooctadeca-trienoic acid

**Table 4 molecules-25-01326-t004:** Influence of drying treatments on kaempferol derivatives in Chinese cabbage and Nightshade leaves.

Chinese Cabbage						Nightshade
Treatments	Kaempferol-sophoroside-hexoside (mg kg^−1^)	Kaempferol-3-*O*-hydroxyferoyl-trihexoside (mg kg^−1^)	Kaempferol-3-*O*-hydroxyferoyl-diglucoside (mg kg^−^^1^)	Kaempferol-dihexoside (mg kg^−1^)	Kaempferol-3-*O*-sinapoyldihexoside-hexoside (mg kg^−^^1^)	Kaempferol 3-*O*-rutinoside (mg kg^−1^)
Raw leaves	0.67 ± 0.07 ^d^	4.10 ± 2.77 ^d^	1.23 ± 0.67 ^f^	0.89 ± 0.72 ^e^	0.45 ± 0.84 ^d^	5.60 ± 0.54 ^d^
Solar drying	391.96 ± 1.36 ^a^	483.94 ± 1.47 ^a^	91.56 ± 0.07 ^b^	147.42 ± 0.16 ^b^	188.96 ± 1.97 ^a^	572.96 ± 0.8 ^a^
Microwave drying	380.44 ± 1.57 ^a^	454.89 ± 1.11 ^b^	63.43 ± 1.22 ^c^	140.87 ± 0.29 ^b^	171.36 + 0.31 ^b^	107.99 ± 0.04 ^d^
Freeze-drying	344.93 + 1.03 ^b^	414.56 ± 1.08 ^c^	55.60 ± 1.19 ^d^	107.7 ± 0.04 ^c^	156.65 ± 0.24 ^c^	533.14b ± 0.56 ^b^
Oven drying	242.06 ± 0.97 ^c^	419.06 ± 0.93 ^c^	96.23 ± 0.83 ^a^	154.54 ± 0.88 ^a^	160.99 ± 0.80 ^c^	397.59 ± 0.75 ^c^
Sun drying	269.45 ± 0.29 ^c^	392.61 ± 1.09 ^c^	20.79 ± 1.61 ^e^	87.44 ± 0.08 ^d^	150.33 ± 0.74 ^c^	30.3 ± 0.05 ^c^

Means followed by the same letter within the column are not significantly different at *p* < 0.05 level.

**Table 5 molecules-25-01326-t005:** Influence of drying treatments on isorhamantin glycosides and sinapoyl malate in Chinese cabbage.

Chinese Cabbage			
Drying Treatments	Isorhamantin -O-dihexoside (mg kg^−1^)	Isorhamnetin-O-hexoside (mg kg^−1^)	Sinapoyl malate (mg kg^−1^)
Raw leaves	0.13 ± 0.02 ^d^	2.34 ± 0.60 ^e^	0.04 ± 0.07 ^e^
Solar drying	222.05 ± 0.25 ^a^	153.42 ± 0.35 ^a^	1413.39 ± 0.53 ^a^
Microwave drying	184.25 ± 0.44 ^b^	42.45 ± 03.11 ^c,d^	909.25 ± 0.94 ^c^
Freeze-drying	184.25 ± 0.44 ^b^	36.45 ± 0.46 ^d^	884.11 ± 0.49 ^c,d^
Oven drying	191.80 ± 0.8 ^b^	117.75 ± 0.56 ^b^	1225.44 ± 0.61 ^b^
Sun drying	167.40 ± 0.68 ^c^	53.78 ± 0.36 ^c^	710.59 ± 0.99 ^d^

Means followed by the same letter within the column are not significantly different, at *p* < 0.05 level.

**Table 6 molecules-25-01326-t006:** Influence of drying treatments on flavonoid glycosides and hydro cinnamic acid esters in Nightshade.

Drying Treatments	Isorhamnetin-*O*-hexoside (mg kg^−1^)	Rutin (mg kg^−1^)	Caffeoylmalic Acid (mg kg^−1^)	Neochlorogenic Acid (mg kg^−1^)	Chlorogenic Acid (mg kg^−1^)
Raw leaf	1.32 ± 0.14 ^d^	2497.04 ± 0.23 ^a^	900 ± 0.45 ^d^	80.00 ± 0.05 ^d^	380.00 ± 0.56 ^b^
Solar drying	14.16 ± 1.11 ^b^	2081.46 ± 1.09 ^d^	2409.59 ± 0.41 ^b^	446.99 ± 0.98 ^a^	614.45 ± 1.08 ^a^
Microwave drying	7.46 ± 1.11 ^c^	2497.04 ± 0.64 ^a^	140.40 ± 0.46 ^e^	8.84 ± 0.42 ^e^	245.52 ± 6.01 ^c^
Freeze-drying	13.89 ± 0.87 ^b^	2403.79 ± 1.06 ^b^	1807.42 ± 0.05 ^c^	129.38 ± 0.65 ^c^	33.18 ± 0.47 ^d^
Oven drying	13.47 ± 0.51 ^b^	2340.49 ± 1.41 ^c^	2901.41 ± 002 ^a^	479.86 ± 0.83 ^a^	598.61 ± 0.84 ^a^
Sun drying	16.19 ± 1.34 ^a^	731.62 ± 1.51 ^e^	2193.81 ± 0.10 ^b^	275.58 ± 1.01 ^b^	380.14 ± 0.10 ^b^

Means followed by the same letter within the column are not significantly different at *p* < 0.05 level.

**Table 7 molecules-25-01326-t007:** Exact Mass/Retention Time pairs responsible for the separation of sundried Chinese cabbage and other samples that underwent solar, oven, microwave and freeze-drying treatments.

Tentative Identification	Retention Time	Mass	p[1]P	p(corr)[1]P	Factor of Change	Sun Drying	Other Drying Treatment
Unknown compound	3.46	325.0555	0.263232	0.774685	2.3	175.826	400.048
Sinapoyl malate	3.98	339.0712	0.385633	0.729198	1.6	800.25	1307.52
Kaempferol-3-O-sophoroside	9.78	601.3231	−0.16337	−0.963747	189.2	73.713	0.389591
Kaempferol-3-sophorotrioside-7-glucoside	2.95	933.2275	−0.19537	−0.678301	1.8	343.176	195.882
Hydroxyoctadecenedioic acid	6.26	327.2169	−0.2870	−0.32450	2.1	234.45	453.20

**Table 8 molecules-25-01326-t008:** Exact Mass/Retention Time pairs responsible for the separation of microwave dried Nightshade leaves and other samples that underwent solar, oven, microwave and freeze-drying treatments.

Primary ID	Retention Time	Mass	p[1]P	p(corr)[1]P	Factor of Change	Microwave Drying	Other Drying Treatment
Hydroxyoctadecanedioic acid	7.46	329.2323	0.275508	−0.986224	20.1	711.09	35.2966
Trihydroxyoctadecadiene derivative	6.36	327.2171	0.322555	−0.929414	4.3	1259.64	291.208
Trihrdoxyoctadecadiene derivative	6.82	327.2166	−0.38672	−0.983541	12.4	116.803	1450.91
Caffeoylmalic acid	3.37	591.0983	0.314166	0.902086	53.2	18.0534	960.365

**Table 9 molecules-25-01326-t009:** Effect of different drying treatments on saponins (Tigogenin-5G and Tigogenin-GG-Rham-Xyl-Xyl.

Treatments	Tigogenin-5G (mg kg^−1^)	Tigogenin-GG-Rham-Xyl-Xyl (mg kg^−1^)
Raw leaves	0.45 ± 0.98 ^e^	0.56 ± 0.34 ^e^
Solar drying	70.54 ± 1.69 ^b^	73.92 ± 2.69 ^d^
Microwave drying	12.36 ± 0.32 ^d^	nd
Freeze-drying	108.68 ± 7.48 ^a^	93.35 ± 2.40 ^a^
Oven drying	59.43 ± 3.11 ^c^	81.57 ± 1.79 ^c^

Means followed by the same letter within the column are not significantly different at *p* < 0.05 level. Nd- not detected.

**Table 10 molecules-25-01326-t010:** Effect of drying treatments on ascorbic acid content in Chinese cabbage and Nightshade leaves.

Drying Treatments	Ascorbic Acid Content (mg 100g^−1^)	
	Chinese Cabbage	Nightshade
Raw leaves	58.1 ^a^ ± 0.34	108.3 ^a^ ± 0.42
Solar drying	47.9 ^b^ ± 012	98.9 ^b^ ± 0.78
Microwave drying	30.5 ^d^ ± 0.65	55.3 ^d^ ± 0.26
Freeze-drying	57.5 ^a^ ± 0.06	107.5 ^a^ ± 0.14
Oven drying	23.6 ^e^ ± 0.18	39.2 ^e^ ± 0.94
Sun drying	40.0 ^c^ ± 0.56	89.1 ^c^ ± 0.13

Means followed by the same letter within the column are not significantly different at p < 0.05 level.

**Table 11 molecules-25-01326-t011:** Effects of different postharvest drying methods on antioxidant activities in Chinese cabbage.

Treatments	FRAP (mg of TEAC100g^−1^)	DPPH (mg of TEAC100g^−1^)	ABTS (mg of TEAC100g^−1^)
Solar drying	4.36 ± 0.00 ^b^	0.18 ± 0.00 ^b^	1.07 ± 0.00 ^c^
Microwave drying	2.94 ± 0.01 ^c^	0.14 ± 0.00 ^d^	0.90 ± 0.00 ^d^
Freeze-drying	4.49 ± 0.00 ^a^	0.28 ± 0.00 ^a^	3.01 ± 0.05 ^a^
Oven drying	1.38 ± 0.00 ^e^	0.11 ± 0.00 ^d^	1.84 ± 0.00 ^b^
Sun drying	2.23 ± 0.01 ^d^	0.12 ± 0.00 ^d^	1.89 ± 0.00 ^b^

Means followed by the same letter within the column are not significantly different at *p* < 0.05 level.

**Table 12 molecules-25-01326-t012:** Effect of different postharvest drying methods on antioxidant activities in Nightshade leaves.

Treatments	FRAP (mg of TEAC100g^−1^)	DPPH (mg of TEAC100g^−1^)	ABTS (mg of TEAC100g^−1^)
Solar drying	3.36 ± 0.05 ^b^	0.27 ± 0.00 ^b^	1.92 ± 0.00 ^a^
Microwave drying	1.52 ± 0.00 ^d^	0.09 ± 0.00 ^d^	1.87 ± 0.00 ^b^
Freeze-drying	3.96 ± 0.02 ^a^	0.30 ± 0.00 ^a^	1.93 ± 0.00 ^a^
Oven drying	1.12 ± 0.01 ^e^	0.03 ± 0.00 ^d^	0.57 ± 0.01 ^c^
Sun drying	2.92 ± 0.04 ^c^	0.25 ± 0.00 ^c^	1.89 ± 0.01 ^b^

Means followed by the same letter within the column are not significantly different at *p* < 0.05 level.

**Table 13 molecules-25-01326-t013:** Proximate composition of solar dried Nightshade and Chinese cabbage leaves.

Proximate Composition	Night Shade	Chinese Cabbage
	g 100g^−1^	g 100g^−1^
Moisture	5.77 ± 0.22	5.51 ± 0.22
Dry matter	94.23 ± 0.34	94.49 ± 0.26
Total fat	1.64 ±1.16	3.46 ± 0.21
Carbohydrates	17.50 ± 1.23	10.57 ± 1.12
Protein	32.91 ± 1.65	33.68 ± 1.68
Total ash	13.37 ± 0.67	16.98 ± 0.85
Total Dietary Fibre (TDF	28.81 ± 1.87	29.80 ± 3.58
Sodium (Na)	289.20 ±1.98	8.89 ± 0.80
Energy	1148.13 kJ	1118.67 kJ

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
