# Peer review of "Effects of Different Drying Methods on Untargeted Phenolic Metabolites, and Antioxidant Activity in Chinese Cabbage (Brassica rapa L. subsp. chinensis) and Nightshade (Solanum retroflexum Dun.)"

_molecules, 2020, doi:10.3390/molecules25061326_

Round 1

Reviewer 1 Report

This paper describes the impact of different drying methods on two leafy greens consumed in Africa.  Areas which need modification or strengthening are:

Italics should be used for scientific names The paragraph from line 45-57 could be reworded and made more concise as it is a bit repetitive The colour change parameters (e.g. a* etc) should be explained References are needed for the statement at line 88 and 113 All of the Tables and the statistical data should be checked as the authors seem to be indicating that p<0.05 is not significant different. It is clear without doing the formal test that values deemed not different in tables are actually different.  Please also use exact p values. P values are denoted differently in different Tables – in some the letter is after the mean and in others after the standard deviation. Overall it is quite hard to distinguish these letters from the number in the Tables. Use of superscripts rather than plain text may assist with clarity. Line 213 – please give data or refer reader to where they can access this information Line 239 – it is unclear why some words are underlined The figures are quite hard to read and I suggest the authors work on improving presentation and readability of these items. In some cases plots are also missing axis labels and text associated with the plot is either overlapping or too small to read. Figure 4 would benefit from better legend labels Table 12 – please include units for each item. Why does one of the energy entries show as kJ/100g and the other not? Why is there a ] at the end of some values? The authors note that the leaves were picked at the 8-leaf stage – is this the normal stage that the leaves are harvested for consumption? If not why was this stage selected and how might the analysis differ for older leaves. Please give more detail of methods used in sections 3.5, 3.7, 3.8, 3.9 There is inconsistent formatting of references in the reference list

Author Response

XXC

Comments and Suggestions for Authors

This paper describes the impact of different drying methods on two leafy greens consumed in Africa.  Areas which need modification or strengthening are:

Many thanks for the constructive comments and we have tried our best to improve the current format and content with your guidance.  

1.   Comment-  Italics should be used for scientific names –

Answer- revised and highlighted in blue font

2.    Comment - The paragraph from line 45-57 could be reworded and made more concise as it is a bit repetitive.

Answer- It has been revised as follows

Since, the growing alertness of functional compounds found in the fruits and vegetables and their beneficial health effects in recent years has pointed to the fact that increased inclusion of fruits and vegetables is essential in our daily diets [5]. Traditional African leafy vegetables are well known for the contribution of both micronutrients and functional compounds to the diets of African consumers. Therefore, traditional African leafy vegetables can be included in diet diversification strategy for the sub-Saharan African population to combat the hidden hunger.

3.     Comment- The colour change parameters (e.g. a* etc) should be explained 

Answer- Explained  as ‘The following colour coordinates were used to express the colour changes. Light intensity (L*), Colour coordinate a* relates to red and green, Colour coordinate b* relates to yellow colour, hº = arc tangent b*/a* represents a basic color like red (~29º), orange (~45º) or yellow (~70º), ∆ E-colour difference, the variation in colour compared to the original sample using the following formulae given in section 3.4.

4.      Comment- References are needed for the statement at line 88 and 113 

Answer- The line 88 and 113 are results do need references.

Revised as On the contrary during hot air (oven) and freeze drying of cabbages (Brassica oleracea L. variety Capitata L.), no differences in brightness (L*) value was reported [12].However, the increase in b* value during postharvest dehydration treatments compared to the fresh leaves observed in Chinese cabbage (Table 1A) and Night shade leaves (Table 1B) are possibly due to the interference of intercellular air trapped on the greenness of the chlorophyll as explained previously by Rajkumar et al. [12] and confirms their findings.

New reference

Rajkumar, G; Shanmugam, S; de Sousa Galvâo, M; Dutra Sandes, R.D;  Leite Neta, M.T.S; Narain, N; Mujumdar, A.S. Comparative evaluation of physical properties and volatiles profile of

cabbages subjected to hot air and freeze drying. LWT Food Technol. 2017, 80, 501-509.

Line 113 the following statement was included in the original submitted version. ‘’ A similar increase in lipophilic pigments during heat treatment was noted in Ipomoea aquatica Forsk [13] and mustard, mint and spinach greens [14]’’- therefore no revision done here.

5. Comment-     All of the Tables and the statistical data should be checked as the authors seem to be indicating that p<0.05 is not significant different. It is clear without doing the formal test that values deemed not different in tables are actually different.  Please also use exact p values. P values are denoted differently in different Tables – in some the letter is after the mean and in others after the standard deviation. Overall it is quite hard to distinguish these letters from the number in the Tables. Use of superscripts rather than plain text may assist with clarity.

Answer- It has been corrected Means followed by the same letter within the column are not significantly different, P< 0.05. We tried to use superscripts, but due to the small font size it not clear to read. Therefore, no change has been done.

6.   Comment-    Line 213 – please give data or refer reader to where they can access this information Line

Answer- Mentioned as ‘Raw Chinese cabbage leaves harvested in summer contained 2.34 mg kg-1 as stated in Table 5.’

7.Comment-      239 – it is unclear why some words are underlined – it is not underlined during submission. May be during the formatting the underlining could have happed.

r- The figures are quite hard to read and I suggest the authors work on improving presentation and readability of these items.

Comment- We do understand the reviewers comment but these images are generated by the soft wear used and we do not have any control over that. But in order to improve the figures more readable we have enlarged the figures.

Comment- In some cases plots are also missing axis labels and text associated with the plot is either overlapping or too small to read. Figure 4 would benefit from better legend labels 

Answer- Figure 4A. Histogram illustrating quantitative differentiation of biomarkers. Generally it is given like in Fig 4. The Y axis illustrates the peak intensity in counts sec -1. Figures have been revised and enlarged. As we have stated above these figures are generated by the software. .  

Comment- Table 12 – please include units for each item.

Answer- All the units have been inserted

 Comment- Why does one of the energy entries show as kJ/100g and the other not? Why is there a ] at the end of some values? 

Answer- All the units are inserted and corrected

Comment- The authors note that the leaves were picked at the 8-leaf stage – is this the normal stage that the leaves are harvested for consumption? If not why was this stage selected and how might the analysis differ for older leaves. Please give more detail of methods used in sections 3.5, 3.7, 3.8, 3.9 There is inconsistent formatting of references in the reference list 

Answer- 8-leaf stage is traditionally used as maturity for harvesting and we are busy with the analyses of harvesting different stages of maturity.

Most of the methods are published by our research group. On that note we cannot give the details of the same methods and the similar report index will become very high. Therefore, we have decided to give the references. Rewriting the methods elaborately can increase the similarity index. Generally, if the methods are published in detail if there are any modifications it has to be mentioned and if not we can refer to the previously published methods.

Comments and Suggestions for Authors

This paper describes the impact of different drying methods on two leafy greens consumed in Africa.  Areas which need modification or strengthening are:

1.     Italics should be used for scientific names – revised and highlighted in blue font

2.     The paragraph from line 45-57 could be reworded and made more concise as it is a bit repetitive.

It has been revised as follows

Since, the growing alertness of functional compounds found in the fruits and vegetables and their beneficial health effects in recent years has pointed to the fact that increased inclusion of fruits and vegetables is essential in our daily diets [5]. Traditional African leafy vegetables are well known for the contribution of both micronutrients and functional compounds to the diets of African consumers. Therefore, traditional African leafy vegetables can be included in diet diversification strategy for the sub-Saharan African population to combat the hidden hunger.

3.     The colour change parameters (e.g. a* etc) should be explained 

Explained  as ‘The following colour coordinates were used to express the colour changes. Light intensity (L*), Colour coordinate a* relates to red and green, Colour coordinate b* relates to yellow colour, hº = arc tangent b*/a* represents a basic color like red (~29º), orange (~45º) or yellow (~70º), ∆ E-colour difference, the variation in colour compared to the original sample using the following formulae given in section 3.4.

4.      References are needed for the statement at line 88 and 113 

The line 88 and 113 are results do need references.

Revised as On the contrary during hot air (oven) and freeze drying of cabbages (Brassica oleracea L. variety Capitata L.), no differences in brightness (L*) value was reported [12].However, the increase in b* value during postharvest dehydration treatments compared to the fresh leaves observed in Chinese cabbage (Table 1A) and Night shade leaves (Table 1B) are possibly due to the interference of intercellular air trapped on the greenness of the chlorophyll as explained previously by Rajkumar et al. [12] and confirms their findings.

New reference

Rajkumar, G; Shanmugam, S; de Sousa Galvâo, M; Dutra Sandes, R.D;  Leite Neta, M.T.S; Narain, N; Mujumdar, A.S. Comparative evaluation of physical properties and volatiles profile of

cabbages subjected to hot air and freeze drying. LWT Food Technol. 2017, 80, 501-509.

Line 113 the following statement was included in the original submitted version. ‘’ A similar increase in lipophilic pigments during heat treatment was noted in Ipomoea aquatica Forsk [13] and mustard, mint and spinach greens [14]’’- therefore no revision done here.

Comment- 5.     All of the Tables and the statistical data should be checked as the authors seem to be indicating that p<0.05 is not significant different. It is clear without doing the formal test that values deemed not different in tables are actually different.  Please also use exact p values. P values are denoted differently in different Tables – in some the letter is after the mean and in others after the standard deviation. Overall it is quite hard to distinguish these letters from the number in the Tables. Use of superscripts rather than plain text may assist with clarity.

Answer- It has been corrected Means followed by the same letter within the column are not significantly different, P< 0.05. We tried to use superscripts, but due to the small font size it not clear to read. Therefore, no change has been done.

Comment- 6.      Line 213 – please give data or refer reader to where they can access this information Line

Answer- Mentioned as ‘Raw Chinese cabbage leaves harvested in summer contained 2.34 mg kg-1 as stated in Table 5.’

Comment- 7.      239 – it is unclear why some words are underlined –

Answer- it is not underlined during submission. May be during the formatting the underlining could have happend.

Comment- The figures are quite hard to read and I suggest the authors work on improving presentation and readability of these items.

Answer- We do understand the reviewers comment but these images are generated by the soft wear used and we do not have any control over that. But in order to improve the figures more readable we have enlarged the figures.

Comment- In some cases plots are also missing axis labels and text associated with the plot is either overlapping or too small to read. Figure 4 would benefit from better legend labels 

Answer- Figure 4A. Histogram illustrating quantitative differentiation of biomarkers. Generally it is given like in Fig 4. The Y axis illustrates the peak intensity counts Sec -1 . Figures have been revised and enlarged. As we have stated above these figures are generated by the software. .  

Comment- Table 12 – please include units for each item.

Answer- All the units have been inserted

 Comment- Why does one of the energy entries show as kJ/100g and the other not? Why is there a ] at the end of some values? 

Answer- All the units are inserted and corrected

Comment- The authors note that the leaves were picked at the 8-leaf stage – is this the normal stage that the leaves are harvested for consumption? If not why was this stage selected and how might the analysis differ for older leaves. Please give more detail of methods used in sections 3.5, 3.7, 3.8, 3.9 There is inconsistent formatting of references in the reference list 

Answer- 8-leaf stage is traditionally used as maturity for harvesting and we are busy with the analyses of harvesting different stages of maturity.

Most of the methods are published by our research group. On that note we cannot give the details of the same methods and the similar report index will become very high. Therefore, we have decided to give the references. Rewriting the methods elaborately can increase the similarity index. Generally, if the methods are published in detail if there are any modifications it has to be mentioned and if not we can refer to the previously published methods.

Most of the methods are published by our research group. On that note we cannot give the details of the same methods and the similar report index will become very high. Therefore, we have decided to give the references. Rewriting the methods elaborately can increase the similarity index. Generally, if the methods are published in detail if there are any modifications it has to be mentioned and if not we can refer to the previously published methods.

The comment regarding the methods. The methods are elaborated in detail in

59.          Ndou A.; Tinyani, P.P.; Slabbert, R.M.; Sultanbawa, Y.; Sivakuma D. An integrated approach for harvesting Natal plum (Carissa macrocarpa) for quality and functional compounds  related to maturity stage. Food chem. 2019. 293,499-510

60.          Mpai, S.; du Preez, R.; Sultanbawa, Y.; Sivakumar, D. Phytochemicals and nutritional composition in accessions of Kei-apple (Dovyalis caffra): Southern African indigenous fruit. Food Chem. 2018. 253, 37-45

1.            Mampholo, B.M.; Sivakumar, D.; Beukes, M.; van Rensburg, JW. Effect of modified atmosphere packaging on the quality and bioactive compounds of Chinese cabbage (Brasicca rapa L. ssp. chinensis). J Sci Food Agric. 2013, 93, 2008–2015

3.            Managa, G.; Remize, F.; Sivakumar, D. Effect of moist cooking blanching on colour, phenolic metabolites and glucosinolate content in Chinese cabbage (Brassica rapa L. subsp. chinensis). Foods 2019, 8, 399

Reviewer 2 Report

Comments to the manuscript molecules-720812 “Effects of different drying methods on phenolic metabolites, and antioxidant activity in Chinese cabbage (Brassica rapa L. subsp. chinensis) and Nightshade (Solanum retroflexum Dun.)”.

The manuscript is the report of an experiment of drying of Chinese cabbage and Nightshade leaves with different techniques (conventional oven drying, solar cabinet drying, sun drying and freeze drying). Colour properties, antioxidant components, chlorophyll content, ascorbic acid, and phenolic components of dried leaves have been determined as compared with the fresh products.

The experiment was correctly designed and the manuscript well organized. The introduction is well written and with a complete state of the art. The research objectives are clearly presented. Results and discussion are sufficiently well presented and the materials and methods complete, as well as the bibliography.

In my opinion, the manuscript may represent a good scientific contribute to the field of research and is suitable for publication after some minor editorial changes.

Please check the following suggestions for minor editorial changes.

1) please check all Tables for format and lettering; in some tables the letters of mean separation label the means and in others are reported after the standard deviation;

2) page 1, line 36 and all over the text: please change “… g-1 FW …” with “… g-1 FW …”;

3) page 1, line 42 and all over the text: please change “… g-1 …” with “… g-1 …”;

4) page 3, line 114 and all over the text: please change “… 100°C …” with “… 100 °C …”;

5) page 4: please check the editing of Tables 2A and 2B;

6) page 9: please check the editing of Tables 4 and 6;

7) page 10 and 13: please check the lettering of the Figures 2 and 4 that are out of the order.

Author Response

Many thanks for the valuable comments and suggestions and we appreciate your time. 

Please check the following suggestions for minor editorial changes.

Comment -1) please check all Tables for format and lettering; in some tables the letters of mean separation label the means and in others are reported after the standard deviation-

Answer- corrected;

Comment- 2) page 1, line 36 and all over the text: please change “… g-1 FW …” with “… g-1 FW …”;

Answer- corrected

Comment- 3) page 1, line 42 and all over the text: please change “… g-1 …” with “… g-1 …

Answer- …”;corrected”;

Comment- 4) page 3, line 114 and all over the text: please change “… 100°C …” with “… 100 °C …”;……”;

Answer- corrected”;

Comment- 5) page 4: please check the editing of Tables 2A and 2B;

Answer- corrected”;

Comment- 6) page 9: please check the editing of Tables 4 and 6;

Answer- corrected”;

Comment- 7) page 10 and 13: please check the lettering of the Figures 2 and 4 that are out of the order.

Answer- As stated previously these figures are generated by the software and we do have any control. But we have tried our best to make this clear for the readers  

Reviewer 3 Report

The manuscript aims to characterize the extracts of two leafy vegetables consumed in Africa, Chinese cabbage and Nightshade. Authors focused on the post-harvest conditions of these herbs and the content of different phytochemicals and antioxidant capacity.

I think the title of the manuscript is not accurate for the showed data. In fact, there are no phenolic metabolites (which usually are the ones appearing in the bloodstream after consumption of vegetables), there are also chlorophyll, and carotenoids and saponins, and ascorbic acid and there is also the carotenoid content.

My first and important concern is about glucosinolates: why they are missing, above all for Chinese cabbage, which might account much more than both phenolics and carotenoids?

The manuscript is not easy to read due to the huge amount of data, and moreover PCA and PLSDA data are difficult to interpret. Maybe it is possible to substitute RT and mass with compounds name where possible.

Author Response

Comments and Suggestions for Authors

Many thanks for the comments and it stimulates and challenges our thinking . Also appreciate you time.  

Comment- The manuscript aims to characterize the extracts of two leafy vegetables consumed in Africa, Chinese cabbage and Nightshade. Authors focused on the post-harvest conditions of these herbs and the content of different phytochemicals and antioxidant capacity.

I think the title of the manuscript is not accurate for the showed data. In fact, there are no phenolic metabolites (which usually are the ones appearing in the bloodstream after consumption of vegetables), there are also chlorophyll, and carotenoids and saponins, and ascorbic acid and there is also the carotenoid content.

Answer- We understand the reviewers concern another student is busy with the feeding trials and sampling the blood to see the metabolites. But we strongly feel this study is important to see the changes in phenolic metabolites and this corresponds well with the special issue theme. However, we need to include the chlorophyll and the colour changes in terms of food science perspective to relate to the natural colour that will link to the sensory panel. We have not included the sensory panel data here again it is not the objective of the special issue.

In addition, it is important to know the changes in the phenolic metabolites because the flavonoid glycosides are bio transformed in the gut to aglycons for intestinal absorption. Also, for the next step on bioavailability and accessibility for the invitro digestion it is important to know the phenolic metabolites.   

We do understand the concern about the glucosinolates but the special issue theme is  ‘’Bioactive Phenolic and Polyphenolic Compounds. On this note priority was not given to glucosinolates’.’   

Round 2

Reviewer 1 Report

I have reviewed the revision and responses of the authors in relation to the comments made on their submission. While some of the comments have been addressed, others still require appropriate responses. The work would also benefit from careful poor reading to ensure correct grammar.

1.       The new text Lines 46-51 requires a reference to provide evidence in support of the statements being made

2.       Reference 5 is used to support the sentence “Since, the growing alertness ….” However the content of reference 5 does not cover this material; rather this reference supports the text in the rest of the new text section

3.       The statement “The purchasing power of consumers is determined by the colour (quality) of the products.” on line 97 requires a reference

4.       The presentation of the inferential statistics in the Tables is confusing and the attribution of p values is incorrect. Comparisons where P<0.05 are statistically significantly different.  Please provide exact p values.

5.       The statement “The green colour of the final product is associated ….” at line 121-123 requires a reference

6.       The statement at line 259-261 (“At the same time, moist cooking treatments in our previous …”) refers to unpublished data – please provide evidence for this statement or remove the reference to unpublished data

7.       While appreciating that the Figures are generated by particular software they are still very difficult to read and in some cases legends are unreadable due to overlapping text.

8.       Table 9 – standard deviation needs to be included with the mean

9.       Table 12 – the authors have not explained why there is a ] at the end of some of the data. If this is a typo then it needs to be removed

10.    Line 555 – what does “…difference (LSD) test with P < 0, ….” Mean?

11.    Sections 3.5, 3.7, 3.8, 3.9 in the methods require more detail. Sufficient detail must be provided such that others can replicate the work. If well-established methods are used a briefer form of the method can be used however there still needs to be more detail than is provided here.  

12.    The reference list needs work to ensure that it matches the journal format – DOIs are missing and there is some inconsistent formatting

Author Response

Reviewer 1

Comments and Suggestions for Authors

I have reviewed the revision and responses of the authors in relation to the comments made on their submission. While some of the comments have been addressed, others still require appropriate responses. The work would also benefit from careful poor reading to ensure correct grammar.

Response - Proof reading was done and we will ask the journal to this as well.

Comment 1

  1. The new text Lines 46-51 requires a reference to provide evidence in support of the statements being made

Response - Inserted as requested

  1. Flyman, M.V.; Afolayan, A.J. The suitability of wild vegetables for alleviating human dietary deficiencies. South African Journal of Botany. 2006, 72, 492-497.
  2. Burchi, F., Fanzo,J., Frison, E. The Role of Food and Nutrition System Approaches in Tackling Hidden Hunger. Int J Environ Res Public Health. 8(2): 358–373 (2011).

Comment 2. Reference 5 is used to support the sentence “Since, the growing alertness ….” However the content of reference 5 does not cover this material; rather this reference supports the text in the rest of the new text section

Response- Revised as Slavin,J.L ., Lloyd, B. Health Benefits of Fruits and Vegetables Adv Nutr. 2012 Jul; 3(4): 506–516..

Comment 3- The statement “The purchasing power of consumers is determined by the colour (quality) of the products.” on line 97 requires a reference

Response- 

  1. Moser, R., Raffaelli, R., Thilmany-McFadden, D. Consumer Preferences for Fruit and Vegetables with Credence-Based Attributes: A Review. Int FoodAgribus Man.14, 121-142 · 2011

Comment 

  1. The presentation of the inferential statistics in the Tables is confusing and the attribution of p values is incorrect. Comparisons where P<0.05 are statistically significantly different.  Please provide exact p values.

Response - Corrected in the revised text. All corrections are highlighted in blue.

Comment. 5 The statement “The green colour of the final product is associated ….” at line 121-123 requires a reference

Response  Reference  included as

[17] Baradey, Y., Hawlader, M.N.A., Ismail, A.F., Hrairi, M. Drying of fruit and vegetables : The impact of different drying methods on product quality. In Ed: Mineo V. Advances in heat pump-assisted drying technology, CRC Press, 239-240

Comment 5- The statement at line 259-261 (“At the same time, moist cooking treatments in our previous …”) refers to unpublished data – please provide evidence for this statement or remove the reference to unpublished data

Response- We removed this statement the article is under review so at this stage we cannot include the reference.

Comment 6-While appreciating that the Figures are generated by particular software they are still very difficult to read and in some cases legends are unreadable due to overlapping text.

Response- We have cleared the figures and the important details are presented now. We kept the previous labels to show that this is an original data generated via softweare.Table 9 – standard deviation needs to be included with the mean

Comment 

Included in the revised text 

  1. Table 12 – the authors have not explained why there is a ] at the end of some of the data. If this is a typo then it needs to be removed

response- It is a typo error. Many thanks for bringing this to our notice

Comment-

  1. Line 555 – what does “…difference (LSD) test with < 0, ….” Mean?

Response It is revised and highlighted in blue font- See the revised text 

Comment.-

  1. Sections 3.5, 3.7, 3.8, 3.9 in the methods require more detail. Sufficient detail must be provided such that others can replicate the work. If well-established methods are used a briefer form of the method can be used however there still needs to be more detail than is provided here.  

Response-

 Method 3.5- given substantially, see the revised text

Method 3.7- given substantially, see the revised text

 Method 3.8- given the important details. This is a standard method- no need to elaborate

Comment 

  1. The reference list needs work to ensure that it matches the journal format – DOIs are missing and there is some inconsistent formatting

Revised

– 

Reviewer 3 Report

Authors replied to referees' queries, but in my opinion not in a satisfactory manner.

I strongly believe the title is not reflecting the manuscript content. And it is really not relevant that authors are performing another study for the phenolic metabolites. In this paper there are no phenolic metabolites (in their future paper they will decide to put or not), so it is not realistic. The choice not to put glucosinolates is due by the Special issue on phenolics. How about carotenoids and saponins present in the manuscript? Sorry, this is not satisfactory. Authors missed to reply to the question about PCA/PLSDA.

Author Response

Reviwer 3

I strongly believe the title is not reflecting the manuscript content. –

Response - We understand that the reviewer wants us to change the title. But the other two reviewers  did not bring out the statement that this reviewer is mentioning   the we disagree with this comment and the title will remain as

Effects of different drying methods Colour properties, Untargeted phenolic metabolites, antioxidant activity in Chinese cabbage (Brassica rapa L. subsp. chinensis) and Nightshade (Solanum retroflexum Dun.)

Comment- And it is really not relevant that authors are performing another study for the phenolic metabolites. In this paper there are no phenolic metabolites (in their future paper they will decide to put or not), so it is not realistic.-

Response- Other two reviewers did not bring out your point

The phenolic metabolites in this paper are as follows (Tables 6 , 7)

Isorhamantin -O-dihexoside, Isorhamnetin-O-hexoside, Sinapoyl malate, Isorhamnetin-O-hexoside, Rutin, Caffeoylmalic acid, Neochlorogenic acid, Chlorogenic acid

Here the methodology adopted is was to show the Untargetted phenolic metabolites .

Identification and quantification of predominant polyphenolic metabolites were performed using the Quadrupole time-of-flight (QTOF) mass spectrometer (MS) UPLC-QTOF/MS (Waters, Milford, MA, USA, as described earlier by Managa et al. [3](2019) and Ndou et al. [58](2019

On this note we prefer the editors take this decision.

 Comment- The choice not to put glucosinolates is due by the Special issue on phenolics. How about carotenoids and saponins present in the manuscript?

Response - Carotenoids are not phenolic compounds and as we stated previously it cannot be included in this issue since the this is a special issue on phenolic compounds

Comment- Sorry, this is not satisfactory. Authors missed to reply to the question about PCA/PLSDA.

The manuscript is not easy to read due to the huge amount of data, and moreover PCA and PLSDA data are difficult to interpret. Maybe it is possible to substitute RT and mass with compounds name where possible.

Response - We have tried to simplify the interpretation of the of the PCA and PLSDA data chemometric data more readable for food scientist.  Some of the compounds are unidentified therefore given in terms of  RT and mass your suggestion was taken into consideration. Please see the revised version ad all the changes are highlighted in blue fonts.